# Experimental Study of As-Cast and Heat-Treated Single-Crystal Ni-Based Superalloy Interface Using TEM

**DOI:** 10.3390/nano13030608

**Published:** 2023-02-02

**Authors:** Runjun He, Miao Li, Xiao Han, Wei Feng, Hongye Zhang, Huimin Xie, Zhanwei Liu

**Affiliations:** 1AECC Beijing Institute of Aeronautical Materials, Beijing 100095, China; 2School of Technology, Beijing Forestry University, Beijing 100083, China; 3School of Aerospace Engineering, Beijing Institute of Technology, Beijing 100081, China; 4AML, Department of Engineering Mechanics, Tsinghua University, Beijing 100084, China

**Keywords:** Ni-based single-crystal superalloy, interface, lattice misfit, TEM

## Abstract

The interface plays an important role in determining strength and toughness in multiphase systems and the accurate measurement of the interface structure in single crystal (SX) Ni-based superalloy is also essential. In this work, the γ and γ′ lattice constant, γ/γ′ interface width at dendritic and interdendritic region of casting and solution treatment SX Ni-based superalloy is measured. Various advanced equipment is used to characterize γ/γ′ interface nanostructure. A typical correlation between interface width and γ/γ′ misfit is also summarized. The interface width in the dendritic region of the as-cast sample is larger than that in the interdendritic region. The misfit in the dendritic region is larger than that in the interdendritic region, which has a trend of negative development. There is a common law of the as-cast interdendritic and dendrite interface sample, where the absolute value of the misfit between the two phases is increasing with the phase interface broadening. The comparison of the as-cast and heat-treated interdendritic sample shows that after heat treatment, the phase interface width increases, the misfit decreases, the lattice constant of γ phase increases, and the lattice constant of the γ′ phase decreases. By comparing the as-cast and heat treated dendrites, the absolute value of the misfit of the as-cast dendrite sample is significantly smaller than that of the heat-treated sample, and the misfit increases with the interface broadening. The comparison between interdendritic and dendritic heat-treated samples shows that the absolute value of the misfit between the two phases is smaller than that of the dendritic as-cast samples, and the absolute value of the misfit also increases with the phase interface broadening. In conclusion, property heat treatment can significantly increase the lattice constants of the γ and γ’ phases, reduce the lattice mismatch at the interface of the two phases, and improve the high temperature stability of the alloy. A better understanding of the microstructure of Ni-based single crystal superalloys will provide guidance for the subsequent design of more advanced nickel-based single-crystal superalloys.

## 1. Introduction

Ni-based superalloys are widely used in the hot sections of aircraft and rocket engines, power-generation turbines, and other challenging environments [1,2,3]. It is this material that offers an unusual combination of excellent physical and chemical properties such as strength, ductility, improved fracture toughness, and fatigue resistance, as well as enhanced creep and oxidation resistance at high temperatures. These alloys are precipitation si0trengthened by coherent ordered Ni3Al (L12) precipitates (γ′ phase) within a face-/8u-*/centered cubic (FCC) Ni-rich matrix (γ phase). Ni-based single-crystal (SX) superalloy can work stably above 1050 °C. Usually, there are two major steps to produce such metal: (1) remelting the master alloy ingot and investment casting, and (2) heat treatment of casting to homogenize the composition of the alloy. The SX superalloy is primarily manufactured by lost-wax casting. Although the crystallographic orientation is always along the [001] direction by putting seed crystal or a selector, the composition distribution is not quite the same inside the casing alloy. These defeats have a negative effect on the alloy’s performance at high temperatures, especially when γ/γ′ eutectic or other segregations are prevalent at the interdendritic.

To enhance the superalloy parts, standard heat treatment must apply to the casting SX superalloys. The standard heat treatment normally contains two steps: solution heat treatment and aging heat treatment. The temperature of solution heat treatment is above 1550 K. It is relatively high because the melting point of the SX is 1600 K. After the solution heat treatment, the coarse γ′ phase, and the other segregation tend to be invisible under an optical microscope, and the difference of the alloy element between dendritic and interdendritic tend to be consistent [4] The critical elements (Re, W, Al, Ti, and Ta) become more uniform as the solution heat treatment time increases from 2 h to 16 h. The distribution coefficients of these elements seem to depend strongly on the solution temperature and time [5]. The lattice constants and misfit values of different SX superalloys are different after full heat treatment. Through the convergent beam electron diffraction (CBED) measurement, the lattice constant in the γ phase varies between 0.3575 nm and 0.3586 nm, and the lattice constant in the γ′ phase varies between 0.3575 nm and 0.3603 nm near the dendrite center [6]. At the same time, the misfits of dendrite and interdendritic regions are (−0.45 ± 0.05)% and 0.1%, respectively. However, in the casting stage, the γ and γ′ phase lattices in the primary dendrite arm (PDA) are 0.3598 nm and 0.3587 nm, respectively. The misfit in PDA changes to (−0.29 ± 0.05)% and it changes to 0.3% in the interdendritic region [7,8]. Studies have shown that the lattice of the γ and γ′ phases strongly depends on the segregation of refractory elements [9,10,11]. The lattice constant and misfit of the γ/γ′ phase are significantly different from those in the casting stage, especially after heat treatment.

During the precipitation of γ′ strengthening phase from γ phase, the mechanisms such as coherent strain strengthening and dislocation cutting are introduced [12]. The two-phase interface between the γ phase and γ′ phase would also affect the mechanical properties of such alloy at the micro-nano level [13,14]. To study the effect of element segregation on the microstructure and mechanical properties of the alloy, many scholars have adopted 3D atomic probe chromatography (APT) to analyze the element content in the two phases [15,16,17,18]. Ge [19] studied the two-phase interface of the second generation SX Ni-based superalloy DD6 by using high-resolution scanning transmission electron microscopy (STEM) images. It was first found that the composition width of the interface was consistent with that of the ordered phase and disordered phase, which was 2.2 nm. S. Meher [20] analyzed the phase interface widths of Co-based superalloy and Ni-based superalloy by combining spherical aberration-corrected STEM with 3D APT, and obtained that the composition width of Co-based superalloy interface and the interface widths of ordered phase and disordered phase was 2.3 nm, and the composition width of Ni-based superalloy interface and the interface widths of ordered phase and disordered phase was 3.3 nm, respectively. Through the verification of 3D APT, it was found that the phase interface width results obtained by the two methods were close. R. Srinivasan [16] studied the phase interface width of Ni-based alloy Rene88 DT by STEM and 3D APT. The results show that there are two different interface widths at the two-phase interface of this alloy; namely, the width of component transformation and the width of ordered-disordered structure transformation. The width of component transformation is 2.7 nm, and the width of ordered–disordered structure transformation is 2 nm, which is obtained by intensity ratio analysis and 3D APT. Zhang [21,22] analyzed the phase interface width of Ni-based SX superalloy by high-resolution transmission electron microscopy (HRTEM) combined with image intensity ratio analysis. The above studies show that the two-phase interface of a Ni-based alloy is not a sudden change but a transitional interface. It was measured experimentally that there are composition transition zones and structure transition zones with a width of about 2–3 nm on the surface between the two phases. The difference in lattice constants near the interface region of the two phases of Ni-based single-crystal superalloy, as well as their different elemental composition, affect key factors such as lattice mismatch and mismatch strains in the two phases, which affect the alloy’s high-temperature mechanical properties and deformation mechanisms on a macroscopic level.

However, these studies are only carried out through such statistics measurements which mainly measure a small simple region and then obtain an averaged result, without directly observing the microstructure and interface of the γ/γ′ phase. Meanwhile, atomic structure diffraction imaging can be interpreted by HRTEM image. As the atomic distribution can be located by distinguishing the dark and bright regions, and the crystal structure or the lattice distortion can be identified by using the geometric phase analysis method to analyze the atomic arrangement module [21,23]. Therefore, at the γ/γ′ interface, due to the obvious difference between the L12-ordered Ni3Al and the FCC structure Ni, the corresponding intensity at the γ, γ′, and γ/γ′ interfaces have different oscillation modes, and the width of γ/γ′ interface can be measured by calculating the gradient length from the γ to γ′ phase [24]. The misfit between the γ phase and γ′ phase of SX Ni-based superalloy directly determines the coherent stress at the two-phase interface, which indirectly affects the stability of the microstructure and the high-temperature mechanical properties of the alloy. At present, the methods used to characterize the lattice misfit between two phases include X-ray diffraction method [25], CBED [26], synchronous diffraction [27], neutron diffraction [28,29], and transmission electron microscopy (TEM) [30]. These methods measure the lattice constant of two phases first and then use the relevant formula to calculate the misfit between the two phases, which is an average value. M. Hoelzel [31] observed the obvious microstructure difference between IN706 alloy and DT706 alloy by TEM and neutron diffraction, and the misfit of IN706 alloy and DT706 alloy was 0.54% and 0.38% by neutron diffraction, indicating that DT706 alloy has higher structural stability. Wang [32] used the X-ray diffraction method to measure the lattice constant and misfit of SX Ni-based superalloys under different states. It was found that the lattice constant was larger than that of a fully heat-treated superalloy, and the absolute value of misfit was larger. After long-term aging treatment, the lattice constant increases, and the absolute value of misfit increases. When measuring Ni-based superalloys with different Re contents at room temperature, it is found that with the increase in Re content, the lattice constants of the γ phase and the γ′ phase increase, and the absolute value of misfit decreases.

In this paper, we measure the γ and γ′ lattice constant, γ/γ′ interface width at dendritic and interdendritic region of casting and solution treatment SX cylindrical sample whose composition is similar to CMSX-4 alloy. To this end, we use optical microscopy (OM), advanced scanning electron microscopy (SEM), HRTEM, and energy-dispersive X-ray spectroscopy (EDS) to characterize the γ/γ′ interface nanostructure. Moreover, a typical correlation between interface width and γ/γ′ misfit has also been summarized. The two-phase interface width in the dendritic region of as-cast SX Ni-based superalloy is larger than that in the interdendritic region, and so is the lattice misfit. A common law between the as-cast interdendritic and dendritic regions is that as the phase interface width increases, so does the absolute value of the misfit between the two phases. After heat treatment, the phase interface width in the interdendritic region increases, the misfit decreases, the phase lattice constant increases, and the phase lattice constant decreases. The absolute value of the misfit between the two phases of the heat-treated interdendritic samples is smaller than that of the dendritic samples, and the absolute value of the misfit between the two phases also increases with the phase interface width. We show that property heat treatment can obviously improve the γ and γ′ lattice constant fitness and reduce the γ/γ′ interface misfit.

## 2. Materials and Methods

The material tested was a second generation SX nickel-based superalloy with a composition similar to that of the CMSX-4 alloy. The alloy used in this paper was created experimentally by melting the parent alloy in a vacuum induction furnace and then solidifying it using directional solidification techniques with a crystal orientation of <001>. The as-cast sample was treated with a solid solution at 1302 °C to dissolve the coarse γ′ phase and the two-phase eutectic structure and then cooled naturally. TEM samples are prepared by the FIB double beam system. The technique detail can be found elsewhere [33]. In situ observations were performed using a transmission electron microscope (TEM) at an accelerating voltage of 300 KV with a beam direction of <001>. EDS analysis was also carried out, and the distribution of the elements Al, Co, Cr, Re, Ni, and W in the two phases was obtained in turn, as shown in Figure 1. Adding elements such as Mo, Cr, and Re that tend to segregate to the γ phase in SX Ni-based superalloys results in an increase in the lattice constant of the γ phase, causing a shift towards a negative value in the lattice misfit. When elements prone to segregation of the γ′ phase are added to the alloy, such as Nb, Ti, Ta, and other elements, the lattice constant of the γ′ phase will increase, resulting in the change in lattice misfit value towards the positive value [34].

The rod-shaped single-crystal alloy used experimentally in this paper was prepared by melting the parent alloy in a vacuum induction furnace, followed by a directional solidification technique with a crystal orientation of <001>. After the directional solidification, the selected observation area was etched with a copper sulfate mixture for a typical time of 10 s. The optical micrograph of the etched dendrites is shown in Figure 2a. The TEM sample preparation in this experiment is simply shown in Figure 2b–f. An area of interest (AOI) in the dendrite/interdendritic was selected in the as-cast or solution heat treated-sample (see Figure 2b,c). A layer of Pt with a thickness of about 2 um was deposited in the AOI, as shown in Figure 2d. Then we cut the sample and fixed it on the bracket, as shown in Figure 2e. Finally, the sample is thinned, and the sample after thinning is shown in Figure 2f. The main purpose of depositing a Pt layer is to reduce the damage to the sample during the thinning process of FIB. The selected region was cut and separated from the sample by using the FIB with an acceleration voltage of 30 kV. At a specific level, the sample is fixed on a V-shaped holder and finally, the specific area of the sample is subjected to ion thinning. When the samples were fixed by the ion gun, attention was paid to preventing the implantation of Ga^+^ ions into the samples. Finally, the sample was thinned with an ion beam with an acceleration voltage of 2 kV, and the thickness of the thinned area is less than 100 nm.

In this work, an intensity ratio analysis method [22,35] is used to calculate the interface width of the two-phase interface. In the high-resolution TEM images, a region with clearly arranged atoms and containing the γ phase and γ′ phase is selected and the intensities of the γ and γ′ phase atomic columns are calculated separately. Due to the alternation of Ni atoms in the γ phase and heavy and light atoms in the γ′ phase, the intensity curve of the γ phase tends to stabilize, whereas the intensity curve of the γ′ phase has alternating peaks and valleys (zigzag distribution). There will be a transition region where the curve is stable and varies greatly, so we can determine the position of the two-phase interface. The amplitude of the zigzag distribution becomes smaller in the transition zone. Therefore, the width of the interface can be determined.

## 3. Results and Discussion

### 3.1. Dendritic in AS-Cast Sample

Figure 3a shows an HRTEM image of the interdendritic region of the as-cast SX Ni-based superalloy in the direction of [001] direction, with the inset diffractograms indicating the disordered γ phase and the ordered γ′ phase. Figure 3b depicts the image of the γ and γ′ phase after filtering and amplification in the red frame area in Figure 3a. The atomic structure is clear. Because the constituent elements of the γ phase are mainly Ni elements, there is no gray difference. In the γ′ phase, light atom Ni and heavy atom Al/Ta appear alternately. The heavy atom is mainly distributed at the vertex of the FCC structure cell, and the light atom is mainly distributed at the face center of the FCC structure cell. The ordered L12, Ni3Al-based structure of the γ′ phase consists of two distinct sublattices: the Ni sublattice corresponding to the face-center positions and the Al sublattice corresponding to the corner of the cubic unit cell. Therefore, the gray value of the γ′ phase atom in the high-resolution image shows a periodic distribution from light to dark. A schematic diagram of the cell structure of the γ and γ′ phases is shown in Figure 3c.

In order to describe the interface width and observe the transition from γ phase to γ′ phase, Figure 4a, a region ABCD containing γ phase and γ′ phase is intercepted in Figure 3b. Image intensity summation of this region is carried out along the AB direction. In Figure 4b, an intensity curve is drawn from column AB to column CD along the direction of AC, where the vertical axis represents the intensity in the <200> direction. Note that when the phase interface is considered to be a straight line over such a finite distance, the vertical axis is parallel to the phase interface. In Figure 4b, the alternating appearance of light (such as Ni) and heavy (such as Al, Ta, et al.) elements in the γ′ phase leads to different peaks in the intensity curve, where the distribution is almost the same as that in the previous literature [36]. The γ phase is mainly composed of Ni elements, and its intensity curve is relatively flat, without large fluctuation. An obvious transition region from the γ to γ′ phase can be observed at the two-phase interface. If the intensity ratio of adjacent atomic columns is close to 1, it indicates that the region is γ phase. In addition, if the intensity ratio of adjacent atomic columns fluctuates around 1 in a specific range, it indicates that the region is γ′ phase. The horizontal axis in Figure 4b,c represents the number of pixels in the direction of the phase interface, whereas the longitudinal axis represents intensity. In these two subgraphs, we can judge that the left side is γ phase, and the right side is γ′ phase. The transition zone is in the middle. The ratio on the right side, the disordered γ phase, remains essentially constant at a value near 1 since the site occupancies on the Ni and Al sublattices are equal in a random solid solution. In contrast, for the ordered γ′ phase, the ratio on the left alternates between ~1.2 and 0.8 due to differences in site occupancy. The amplitude of the zigzag distribution is becoming much less in the transition areas, as shown in the distance between the red lines. The width of the interface was calculated to be 1.49 nm, based on the difference in site occupancy. In this work, ordered-disordered transition regions in long-range component transitions are hardly identified as they are in the literature [35]. The difference in the interface width between this work and the others [22,35] may be related to the different compositions and different heat treatments.

In modern SX Ni-based superalloys, the lattice constant of the γ phase α*_γ_*_′_ is usually a little smaller than the lattice constant of the γ matrix α*_γ_*, as is the case in most commercial alloys. This gives rise to a negative mismatch *δ*, which is defined as follows.
(1)$$\delta =2\frac{\alpha_{\gamma \prime }-\alpha_{\gamma }}{\alpha_{\gamma \prime }+\alpha_{\gamma }}$$

A subpixel accuracy transform (SAT) method [22] is used to calculate the lattice constant. High-resolution images at the interface need to be captured accurately when the SAT method is used to solve lattice constant. In general, the γ and γ′ phase can be clearly distinguished in the low-magnification TEM images, but it is difficult to directly observe the atomic resolution TEM images to distinguish the two phases. Therefore, it is necessary to combine the real-time FFT transform to obtain the diffraction spectrum to determine the two phases and the interface position.

The SAT method is used to calculate the lattice constants of the two phases in Figure 1a. The lattice constants of the γ phase and γ′ phase are 0.37128 and 0.37018, respectively. The misfit between the two phases is −0.30% calculated by Equation (1). This is slightly different from the misfit of −0.40% of the second generation single crystal alloy in the Reference [37]. The reason for this result may be that the elements contained in the experimental materials are different, resulting in different distribution ratios of elements in the two phases, different lattice constants, and ultimately different misfits [38,39]. One of the purposes of this research is to discuss the relationship or law between the two-phase interface width and the misfit. So, the same method is used to calculate the two-phase lattice constants at different positions of the as-cast interdendritic samples, as well as the two-phase interface width and the misfit.

Table 1 shows that the lattice constant of the γ phase is between 0.36 and 0.37 regardless of the region among the dendrites, which is somewhat different from the lattice constant measured by the X-ray diffraction experiment in the Reference [40], which is between 0.35 and 0.36. This may be caused by different materials. The difference in the element content of the alloy will lead to the change in its microstructure, thereby affecting the change in the lattice constant of the alloy. Because the transformation between the γ and γ′ phase is not a sudden change, but a gradual transition region, it may lead to the different phase interface widths between the two phases in different regions. With the increase in the interface width, the misfit between the two phases has a negative development trend, which makes the phase interface width and the misfit present a negative correlation.

### 3.2. Dendrite in As-Cast Sample

To further study the difference in phase interface width and lattice misfit in different structures of the SX Ni-based superalloy at the micro-nano level, the dendritic part of the as-cast sample was studied here. The calculation method and calculation process are the same as those in Section 3.1.

From Table 2, we know that the lattice constants of the two phases in the dendritic region of the as-cast SX Ni-based superalloy are slightly larger than those in the interdendritic region, and the misfit value also has some changes. The misfit value of the dendritic region is about −0.30%, whereas the misfit value of the interdendritic region is mostly between −0.30% and −0.50%. The width of interface also has a certain change. Compared with the interdendritic region, the width of the phase interface in the dendrite region has a certain increasing trend. When studying the relationship between phase interface width and misfit, it is found that there is a similar rule between as-cast dendrite and the interdendritic region. With the increase in interface width, the absolute value of misfit between the two phases is also increasing. Studies have shown that Re, W, and Co elements are enriched in the dendrite region, whereas Ti, Ta, Mo, Al, and Cr elements are mainly enriched in the interdendritic, and Re and Ti elements are seriously enriched [41]. Among these elements, the Re element is the key element affecting the misfit. The increase in the content of the Re element will lead to the increase in the lattice constant and then lead to the decrease in the absolute value of the misfit, which is consistent with the results in the Reference [42].

### 3.3. Interdendrite in a Solid Solution Heat-Treated Sample

After heat treatment, the microstructure would be different, including the two-phase interface width, and the related effect on misfit. The solution heat-treated interdendritic region was studied here.

Table 3 shows the interface width, misfit, and lattice constant at different positions of solution heat-treated interdendritic samples. The phase interface width of the heat-treated interdendritic samples increases compared with that of the as-cast interdendritic samples, and the misfit value between the two phases shows a certain decreasing trend. The change in the misfit is bound to have a change in the lattice constant. Therefore, compared with the as-cast interdendritic samples, the lattice constant of the γ phase increases, and the lattice constant of the γ′ phase decreases. The possible reason is that heat treatment leads to the change in element segregation between the two phases, which is consistent with the results in the Reference [43]. Heat treatment will promote the diffusion of elements and tends to make the element content in the two phases uniform, which leads to the increase in the lattice constant of the γ phase, the decrease of lattice constant of the γ′ phase, and the decrease in the misfit. It is also found that the interdendritic samples after heat treatment have a similar law to the as-cast interdendritic samples; that is, the absolute value of misfit increases with the increase in the interface width.

### 3.4. Dendrite in a Solid Solution Heat-Treated Sample

As in previous sections, the solution heat-treated dendrite was also studied here to make the experiment more complete and comprehensive.

It can be seen from Table 4 that compared with the as-cast dendritic samples, the lattice constant of the γ phase of the dendritic samples after heat treatment decreases to a certain extent and the misfit between the two phases increases. When the phase interface width is the same or close, the misfit of the dendritic samples after heat treatment is larger. From the reference [44], the content of the Re in the dendrite region increased from 2.76% to 2.88% after solid solution treatment for 1 h. After solution treatment for 20 h, the content of Re element in the dendrite decreased from 2.88% to 2.22%. Re element is the key element to determining the lattice constant and misfit value. The increase in Re element content will lead to the larger lattice constant of the γ phase, which will reduce the misfit value. Therefore, the misfit of the dendrite sample after solution treatment is larger. Zhou [45] used an energy spectrometer and X-ray diffraction technology to analyze the composition and measure the lattice constant of an SX Nil-based superalloy, respectively. The increase in Re content will increase the lattice constant of γ phase and reduce the lattice misfit between the two phases. The yield strength and creep resistance of the alloy have also been improved.

To express the comparison between the four groups of different samples more clearly, the relationship between the interface width and the misfit value is expressed, as shown in Figure 5. Figure 5a shows the comparison between as-cast interdendritic samples and as-cast dendritic samples. The absolute value of the misfit between the two phases of as-cast interdendritic samples is significantly greater than that of the as-cast dendritic samples, and the absolute value of the misfit increases with the increase in the phase interface width. Figure 5b shows the relationship between phase interface width and misfit between the as-cast interdendritic samples and the heat-treated interdendritic samples. The phase interface width of the heat-treated interdendritic samples is significantly larger than that of the as-cast interdendritic samples, and the absolute value of misfit of heat-treated samples is significantly smaller than that of the as-cast samples. From the perspective of the curve trend, the two samples have the same trend, and the absolute value of misfit increases with the increase in phase interface width. Figure 5c is the comparison between the as-cast and heat-treated dendritic samples. The absolute value of the misfit of the as-cast dendritic sample is less than that of the heat-treated sample, but the difference among the misfit values is small, which may be related to the small content of Re in the dendritic area. Following heat treatment, the amount of Re in both the as-cast and heat-treated samples experiences only a slight change, leading to a corresponding alteration in the misfit. Similarly, with the increase in the interface width, the absolute value of the misfit of the two samples increases. Figure 5d is the comparison between the heat-treated interdendritic samples and the dendrite samples. The absolute value of the misfit of the heat-treated interdendritic samples is less than that of the dendrite samples. Because the Re content in the interdendritic is much larger than that in the dendrite, the increase in the Re content will lead to the decrease in the misfit. So, the absolute value of the misfit in the interdendritic is less than that in the dendrite. However, with the increase in the phase interface width, the absolute value of misfit also increases. We can conclude that the absolute value of the misfit of the second generation SX Ni-based superalloy increases with the increase in the phase interface width. As lattice mismatches between the two phases of the alloy introduce mechanisms such as co-lattice strain and dislocation cutting, different degrees of mismatch will correspond to different mechanical properties. C.H. Zenk [46] used the XRD technique to calculate the two-phase lattice constants of cobalt-based alloys and the mismatch degrees and used the relevant equations to calculate the interphase stress, which showed that the stress was 228 MPa at a mismatch degree of 0.28% and 268 MPa at a mismatch degree of 0.52%, whereas the interphase stress decreased when the mismatch degree continued to increase.

## 4. Conclusions

The width of the phase interface between the interdendritic and dendritic regions of nickel-based single crystal high-temperature alloys, as well as the lattice mismatch, have been studied and analyzed experimentally. The main conclusions in this paper are as follows:(1)The two-phase interface width in the dendritic region of as-cast SX Ni-based superalloy is larger than that in the interdendritic region. The misfit in the dendritic region is larger than that in the interdendritic region, which has a negative development trend;(2)There is a common law between the as-cast interdendritic and dendritic regions, which states that as the phase interface width increases, so does the absolute value of the misfit between the two phases;(3)The comparison of as-cast and heat-treated interdendritic samples reveals that after heat treatment, the phase interface width in the interdendritic region increases, the misfit decreases, the phase lattice constant increases, and the phase lattice constant decreases. By comparing the as-cast and heat-treated dendrites, it is found that the absolute value of the misfit of the as-cast dendrite sample is significantly smaller than that of the heat-treated sample, and the misfit between the two phases increases with the interface width;(4)The comparison between heat-treated interdendritic samples and heat-treated dendritic samples showed that the absolute value of the misfit between the two phases of the heat-treated interdendritic samples is smaller than that of the dendritic samples, and the absolute value of the misfit between the two phases also increases with the phase interface width.

Calculating the lattice mismatch and interface width of samples in different states will help us better understand the microstructure of Ni-based single-crystal superalloys, have a deeper understanding of the relationship between the lattice mismatch and interface width, and also better understand the impact of the lattice mismatch and interface width on the micro-mechanical properties of Ni-based single-crystal superalloys, which will play a substantive role in the design of the next generation of excellent Ni-based single-crystal superalloys.

## Figures and Tables

**Figure 1 nanomaterials-13-00608-f001:**
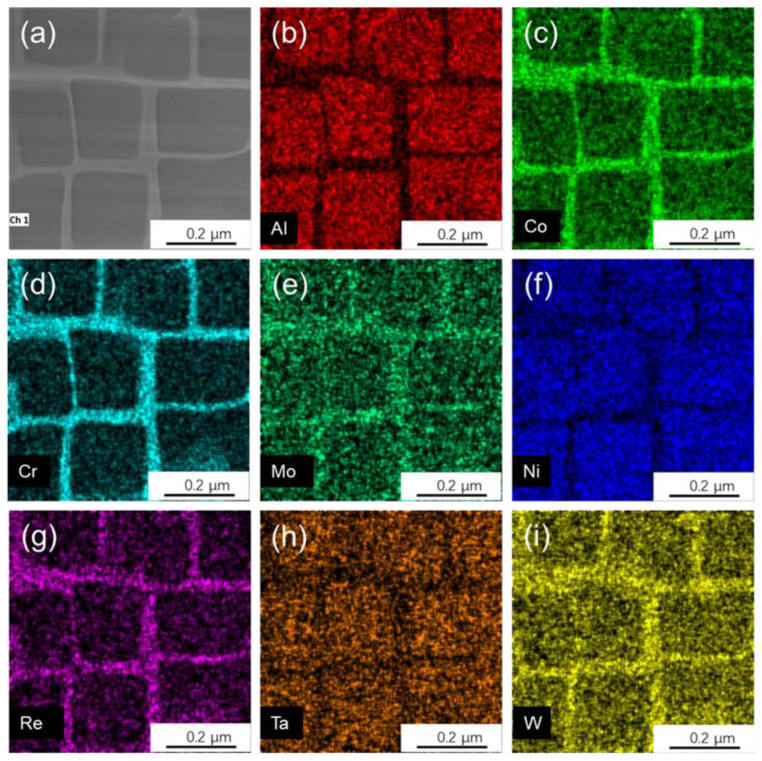
HADDF-EDS mapping of the tested as-cast interdendritic sample. (**a**)No elements. (**b**) Al. (**c**) Co. (**d**) Cr. (**e**) Mo. (**f**) Ni. (**g**) Re. (**h**) Ta. (**i**) W.

**Figure 2 nanomaterials-13-00608-f002:**
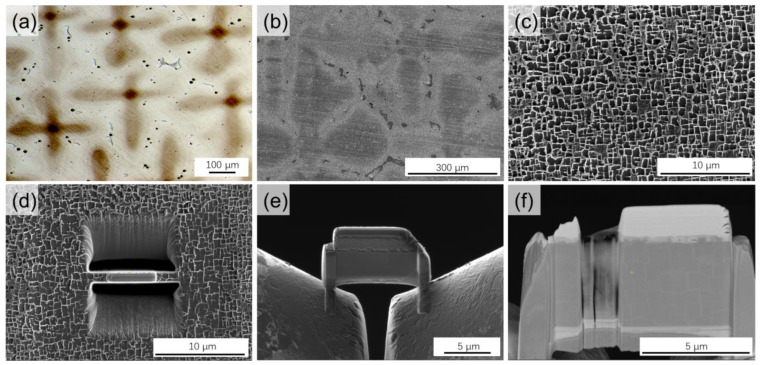
(**a**) Optical images of interdendritic and dendritic regions. (**b**) SEM image of the interdendrite and dendrite regions. (**c**) SEM image in the dendrite region containing the γ and γ′ phase. (**d**) Sample preparation by FIB etching. (**e**) Finished cut sample. (**f**) TEM sample.

**Figure 3 nanomaterials-13-00608-f003:**
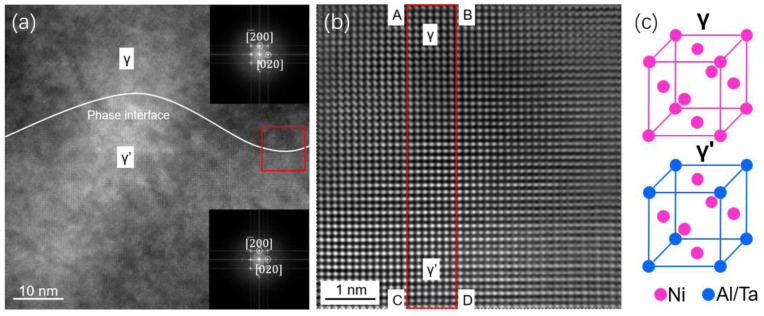
(**a**) HRTEM images of the [001] oriented γ/γ′ phase and of the interface between the two phases. The inset pictures show the Fourier transform images of the γ and γ′ phases. (**b**) Enlarged HRTEM image of the γ/γ′ interface. (**c**) Schematic diagram of the cell of the γ and γ′ phases.

**Figure 4 nanomaterials-13-00608-f004:**
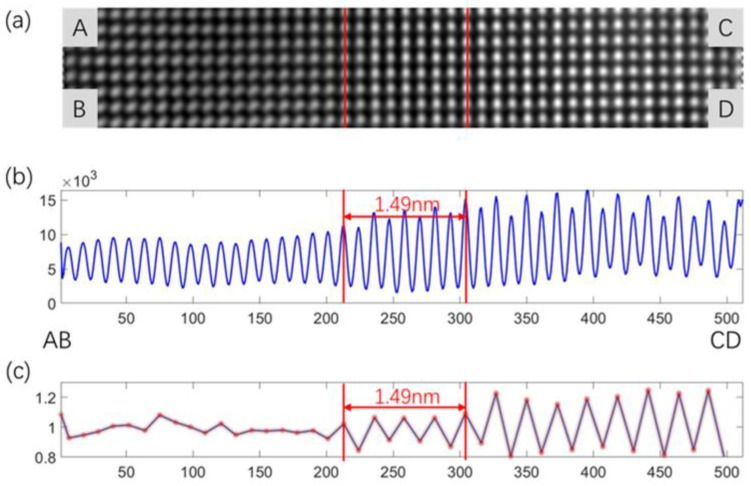
Intensity ratio analysis. (**a**) TEM image of the red boxed area ABCD filtered in Figure 3b. (**b**) Total intensity curve from column AB to column CD showing the transition from ordered γ′ to disordered γ. (**c**) Intensity ratio of each atomic column (peak in b) to the neighboring column on the right.

**Figure 5 nanomaterials-13-00608-f005:**
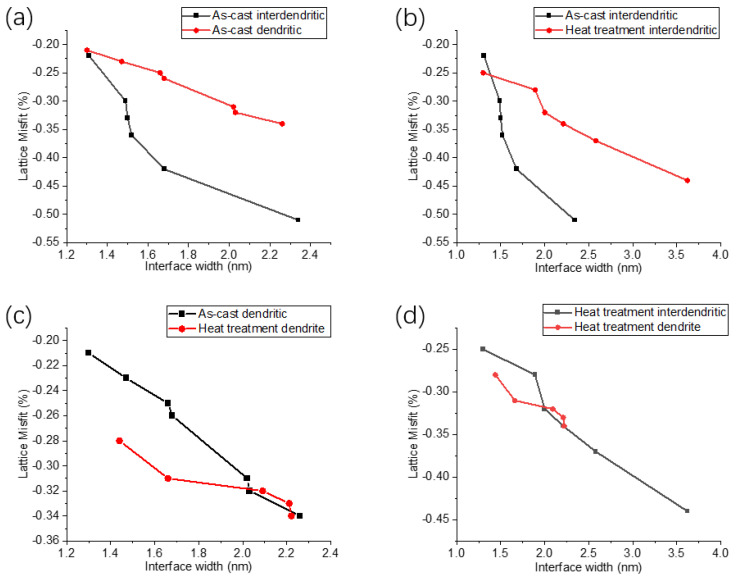
Relationship between phase interface width and misfit: (**a**) Interdendritic and dendrite in as-cast sample; (**b**) Dendrite in as-cast and solid heat-treated sample; (**c**) Interdendritic in as-cast and solid heat-treated sample; (**d**) Interdendritic and dendrite in solid solution heat-treated sample.

**Table 1 nanomaterials-13-00608-t001:** Interface width, misfit, and lattice constant at different positions of as-cast interdendritic samples.

No.	Interface Width	Misfit	Lattice Constant (γ)	Lattice Constant (γ’)
1	2.34 nm	−0.51%	0.36897	0.36708
2	1.68 nm	−0.42%	0.37213	0.37056
3	1.49 nm	−0.30%	0.37128	0.37018
4	1.31 nm	−0.22%	0.36856	0.36776
5	1.50 nm	−0.33%	0.37152	0.37030
6	1.52 nm	−0.36%	0.37225	0.37090

**Table 2 nanomaterials-13-00608-t002:** Interface width, misfit, and lattice constant at different positions of as-cast dendrite samples.

No.	Interface Width	Misfit	Lattice Constant (γ)	Lattice Constant (γ’)
1	2.26 nm	−0.34%	0.37514	0.37386
2	1.47 nm	−0.23%	0.36716	0.36629
3	2.02 nm	−0.31%	0.36710	0.36598
4	1.66 nm	−0.25%	0.37180	0.37088
5	1.30 nm	−0.21%	0.36974	0.36896
6	1.68 nm	−0.26%	0.37070	0.36976
7	2.03 nm	−0.32%	0.37132	0.37014

**Table 3 nanomaterials-13-00608-t003:** Interface width, misfit, and lattice constant at different positions of solution heat-treated interdendritic samples.

No.	Interface Width	Misfit	Lattice Constant (γ)	Lattice Constant (γ’)
1	2.58 nm	−0.37%	0.36952	0.36816
2	2.21 nm	−0.34%	0.36948	0.36824
3	1.89 nm	−0.28%	0.36273	0.36172
4	1.30 nm	−0.25%	0.37098	0.37006
5	2.00 nm	−0.32%	0.36743	0.36624
6	3.62 nm	−0.44%	0.37184	0.37018

**Table 4 nanomaterials-13-00608-t004:** Interface width, misfit, and lattice constant at different positions of solution heat-treated dendrite samples.

No.	Interface Width	Misfit	Lattice Constant (γ)	Lattice Constant (γ’)
1	2.22 nm	−0.34%	0.37096	0.36972
2	1.44 nm	−0.28%	0.36458	0.36356
3	2.21 nm	−0.33%	0.37262	0.37142
4	2.09 nm	−0.32%	0.37274	0.37158
5	1.66 nm	−0.31%	0.37138	0.37024

## Data Availability

Not applicable.

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
