# Peer review of "Experimental Study of As-Cast and Heat-Treated Single-Crystal Ni-Based Superalloy Interface Using TEM"

_nanomaterials, 2023, doi:10.3390/nano13030608_

Round 1

Reviewer 1 Report

The manuscript is well written and structured. It reports valuable research results which are original and of considerable significance to the community. The reviewer recommends it for publication. 

Author Response

Response:
We thank the reviewer for reading our paper carefully and giving the above positive comments.

Reviewer 2 Report

1. The first sentence in the "Abstract", what are the "several properties", please specify.

2. In the Introduction, the effect of γ/γ' interface on the properties of materials can be introduced more, and the specific use range and value of materials should be further explained.

3. Please specify the different elements in Figure 1.

4. Will the Pt thickness adversely affect the material and how to reduce or avoid the effect?

5. Line 321, please specify this will effect what properties of the materials.

Author Response

Please read the attachment.

Reviewer 3 Report

Quite good work. I enjoyed reading it. Major revisions are in order for the authors to address the comments below:

Language needs to be polished. Some minor mistakes found.

“Ni-based superalloys are widely used in the hot section of aircraft and rocket engines, power-generation turbines, and other challenging environments”: see also 10.1016/j.optlastec.2020.106244 and 10.1016/j.addma.2022.102958 and further complement.

“Manufacturing such a high melting point superalloy is not easy.”: why not? Detail please.

“ic, and these defeats hav”: language.

“high-energy synchronous diffraction”: incorrect name. Should be synchrotron. See 10.1016/j.jmrt.2022.08.169 and revise.

Beautiful TEM work!

“the dendrite/interdendritic was selected in the as-cast or s”: can the drendrites be erased by the heat treatment?

“n should be paid to preventing the implantation of Ga+ ions”: and how is this prevented? Did the authors analyzed the composition to search for Ga?

Amazing HRTEM work!

“ntensity curve is relatively flat, without large fluctuation”: what is the significance? Discuss please.

“between ~1.2 and 0.8 due to th”: what drives this? How to control? What is the influence?

“a little smaller than that of the γ matri”: why? Discuss please.

I’m curious have the authors checked the hardness or the mechanical response of the material under different heat treatment conditions? Can this be correlated with the microstructures observed? Clarify please.

“maller than that of the dendritic samples, a”: but was is the significance? Unclear still.

Author Response

Please read the attachment. 

Reviewer 4 Report

The authors have made a comparative study on as-cast and heat treated single-crystal Ni-based superalloy interface using TEM imaging. This work is interesting, but the presentation and language is poor.

1.     The goal of the abstract is unclear. There are several instances of poor writing. For example: “In conclusion, property heat treatment can improve…”.

2.     The authors have used many abbreviations in the introduction section. Please write the originality of this work at the end of Introduction section.

3.     What is “as-cast interdendritic sample”? The authors should improve the caption.

4.     Please check the heading “Solution heat treatment dendrite”. The meaning is ambiguous.

5.     Please explain each of the images in Figure 2. The sub-captions should be provided.

6.     The axes should be labeled in Figure 4.

7.     The errors in measurement should be added in figure 5.

8.     The conclusions should be revised. It is not a summary. Please cite important conclusions in bulleted form.

9.     Additionally, English should be polished. 

Author Response

Please read the attachment. 

Round 2

Reviewer 2 Report

Accept

Reviewer 3 Report

All comments addressed. Acceptance is recommended.